Investigation of the efficacy of siRNA-mediated KRAS gene silencing in pancreatic cancer therapy

Küçükekmekci Büşra 1
Budak Yıldıran Fatma Azize azizebudak@kku.edu.tr 2
1 Institute of Sciences, Department of Biology, Kırıkkale University , Kırıkkale , Yahşihan , Turkey
2 Vocational High School of Health Care Services, Department of Medical Services and Techniques, Kırıkkale University , Kırıkkale , Yahşihan , Turkey
Hromić-Jahjefendić Altijana
Electronic publication date: 2024 Nov 12
Publication date: 2024
Volume: 12
Electronic Location ID: e18214
Received 2024 Feb 6; Accepted 2024 Sep 11
Copyright: ©2024 Küçükekmekci and Budak Yıldıran
Copyright year: 2024
Copyright holder: Küçükekmekci and Budak Yıldıran
License: This is an open access article distributed under the terms of the Creative Commons Attribution License, which permits unrestricted use, distribution, reproduction and adaptation in any medium and for any purpose provided that it is properly attributed. For attribution, the original author(s), title, publication source (PeerJ) and either DOI or URL of the article must be cited.
License URL: https://creativecommons.org/licenses/by/4.0/

Keywords: siRNA, Capan-1, KRAS, Real-Time PCR, AuNP

Funding: Kırıkkale University Scientific Research Projects (BAP) 2021/124 This article was supported by Kırıkkale University Scientific Research Projects (BAP) unit within the scope of project no. 2021/124. The funders had no role in study design, data collection and analysis, decision to publish, or preparation of the manuscript.

==============================
Aim

Pancreatic carcinoma is an aggressive cancer that progresses without many symptoms. The difficulty of early diagnosis and an inadequate response to traditional treatments also cause the survival rate of pancreatic cancer to be low. Current research is focusing on methods of diagnosis and treatment, such as gene therapy, to increase survival rates. Small interfering ribonucleic acid (siRNA) has emerged as a promising advanced therapeutic strategy for cancer treatment. This study sought to silence the KRAS gene in the human pancreatic carcinoma cell line using a complex of small interfering ribonucleic acid (siRNA) and gold nanoparticles (AuNP).

Methods

In this study, 25 nM siRNA and gold nanoparticles at 0.5 mg/ml, 0.25 mg/ml, and 0.125 mg/ml concentrations were used to silence the KRAS gene in the CAPAN-1 cell line. Real-time PCR analysis, agarose gel electrophoresis, and double staining were carried out, and xCelligence real-time cell analysis (RTCA) was used to measure proliferation.

Results

The PCR analysis revealed crossing point (CP) values of actin beta (ACTB) ranging from 33.04 to 35.98, which was in the expected range for all samples. The interaction between the gold nanoparticle/siRNA complex in the double staining analysis revealed that the most effective concentration of gold nanoparticle was 0.125 mg/ml. The WST-1 technique showed that siRNA/AuPEI cells in application groups had a viability rate of over 90%, indicating no toxicity or side effects. The xCELLigence RTCA® showed that at hour 72, there was a significant difference in proliferation in the 0.5 mg/mL PEI/AuNP-siRNA, 0.25 mg/mL PEI/AuNP-siRNA, and 0.125 mg/mL PEI/AuNP-siRNA application groups compared to the control and siRNA groups (p < 0.05). By hour 96, all three groups were statistically different from the control and siRNA groups in terms of proliferation (p < 0.05).

Conclusions

The results of this analysis suggest that the AuPEI/siRNA complex can be effectively used to silence the target gene, but more studies are needed to verify these results.

INTRODUCTION

According to the World Health Organization (WHO), “cancer” refers to a broad group of diseases characterized by the growth of abnormal cells beyond their normal limits, which can spread to various parts of the body, including organs. Cancer has many anatomical and molecular subcategories, each requiring an individual treatment approach. (Pavlopoulou, Spandidos & Michalopoulos, 2015; Ilic & Ilic, 2016; McGuigan et al., 2018; United States Congress, 2007). DNA damage, mutations in proteins responsible for the DNA repair mechanism, epigenetic modifications, mutations in apoptotic pathways, disruptions in RNA metabolism, and errors in cell cycle checkpoints all play important roles in the development of cancer (Zhi et al., 2014; Farazi et al., 2011). There are more than 100 known types of cancer, and each one is an individual disease. In addition to traditional cancer treatments, such as chemotherapy, radiotherapy, and surgery, technological advancements are leading to new treatment methods. Targeted gene therapies are becoming more widely adopted (Pucci, Martinelli & Ciofani, 2019; Zugazagoitia et al., 2016).

The pancreas is a crucial organ located in a retroperitoneal position in the upper part of the stomach between the duodenum and spleen. The pancreas weighs 70–150 g, is 14–20 cm in length, and has both exocrine and endocrine functions. Pancreatic cancer is also called infiltrative pancreatic ductal adenocarcinoma (PDAC; Wolfgang et al., 2013; Hruban et al., 2004). Pancreatic cancer is an aggressive cancer that progresses without many symptoms. The difficulty of early diagnosis and an inadequate response to traditional treatments also cause the survival rate of pancreatic cancer to be low. Current research is focusing on methods of diagnosis and treatment, such as gene therapy, to increase survival rates. Gene therapy is performed by producing the missing protein in the body, increasing the expression of healing proteins, or blocking the production of deleterious proteins (Cooney, Smith & Chung, 2007; Ginn et al., 2018). The gene silencing approach requires antisense, RNA interference, microRNA (miRNA), aptamer, and ribozyme technologies. Gene silencing can occur during transcription or translation (Cooney, Smith & Chung, 2007; Erdmann & Barciszewski, 2012). RNA interference begins with the cutting of double-stranded RNA into small inhibitory RNAs (siRNA) by an RNase III enzyme called Dicer. These RNAs then bind to the RNA-induced silencing complex (RISC), a multiprotein RNA nuclease complex. The RISC uses siRNAs to find complementary mRNA and then cuts the target mRNA in an endonucleolytic manner. Consequently, the decrease in specific mRNAs leads to the decrease in corresponding proteins (Patil, Zhou & Rana, 2014).

The success of these methods depends on the delivery and effective expression of the nucleic acid–based molecule to the target cell. Due to their physical properties, nucleic acid–based molecules encounter various extracellular and intracellular obstacles when circulating in the blood (Giacca, 2010). The first step of gene therapy is transferring the healthy gene into the targeted cell of the patient. This is done by transferring molecules, which are named as vectors. There are two types of gene transferring vectors: biological vectors and physical or chemical vectors. Biological vectors refer to plasmid or viral-mediated systems, and physical or chemical vectors are non-viral vectors that transfer the gene to the cells using chemical or physical pathways (Attar, 2017; Cooney, Smith & Chung, 2007; Scherman, 2014).

Gold nanoparticles (AuNPs) are significant nanomaterials in gene transfers because of their distinct optical, electrical, sensory, and metabolic characteristics. AuNPs are very efficient nanocarriers for a wide range of active compounds, including peptides, proteins, plasmid DNAs (pDNA), siRNAs, and chemotherapeutic drugs. The low immunogenicity of AuNPs makes it possible for them to effectively deliver siRNA, a strong gene inhibitor, to cells within the body, promoting the clinical use of siRNA therapy for genetic diseases (Dykman & Khlebtsov, 2012; Singh et al., 2018; Huang et al., 2007; Elahi, Kamali & Baghersad, 2018; Westcott et al., 1998; Kong et al., 2017).

Many studies show that AuNPs, modified by components such as polyethylene (bPEI), are efficient and safe intra-cellular delivery carriers for siRNA. Shaat et al. (2016) applied siRNA, both alone and with AuNP and bPEI/AuNP, to silence the c-Myc gene. The effective delivery of siRNA using bPEI/AuNP was reported. They also reported that the bPEI/AuNP complex increased the cellular uptake of siRNA without significant cytotoxicity, indicating this complex was both safe and effective.

In this study, cationic bPEI-coated AuNPs were used for intracellular siRNA delivery targeting the KRAS gene in human pancreatic cancer cells (CAPAN-1). AuNPs were synthesized using the modified laser ablation (LAL) method in liquid (Xu et al., 2014) and coated with bPEI, a cross-linking agent that imparts a bPEI positive charge. The silencing efficacy of both siRNA and the siRNA/bPEI/AuNP complex was validated by investigating KRAS gene expression.

Materials and Methods

Materials

The CAPAN-1 human pancreatic cancer cell line was cultured in RPMI-1640 media, and a double staining solution was prepared using ribonuclease A, Hoechst 33342, and propidium iodide. Roche’s High Pure RNA Isolation Kit was used for RNA isolation, and the Transcriptor High Fidelity cDNA Synthesis Kit was used for cDNA synthesis. The LightCycler® 480 system was used to examine the expression of the KRAS gene. Housekeeping GADPH, KRAS R5′-TTGGATCATATTCGTCCACAA-3′, and KRAS F5′-ACTTGTGGTAGTTGGAGCAGA-3′ primers were used in the RT-PCR analysis. The siRNA used in this study was obtained from Applied Biological Materials Inc. (Cat. No. 25910171). Gold nanoparticles, which are polymeric nanoparticles, were covered with polyethylene imine to ensure the controlled transfer of siRNA into the target gene.

siRNA stock

The target gene was silenced using siRNAs, including KRAS-Homo-643 R-GGUGUUGAU UGCCUUCUTT-AGAAGGCAUCAACCTT, Homo-585 F-GGACUUAGCAAGUUAUTT R-AUAACUUCUUGCUAAGUCCTT, and Homo-244 F-GCCUUGACAGCUAATT R-UUAGCUGUAUCGUCAAGGTT. For this study, 2.5 nmol (1 OD) of oligo siRNA was diluted with 125 µl of DEPC water for a final concentration of 20 µM. Three different concentrations of siRNA were tested: 25 nM, 50 nM, and 100 nM. The results showed that the most appropriate siRNA concentration was 25 nM.

Cell culture

This study used CAPAN-1 cells (ATCC-HTB-79TM) that had undergone cryopreservation and subsequent thawing. CAPAN-1 pancreatic cancer cells were put in vials containing DMEM supplemented with L-glutamine. RPMI-1640 medium with 1 ml of 10% FBS and 1% antibiotic was then added. The cell pellet was then placed in a 37 °C incubator with 5% CO2 for proliferation. One day before the application, 48-well plates were prepared with 5x103 cells. Cell proliferation was monitored every 24 h. After completing the surface coating, the cultures were treated with trypsin and transferred to new flasks. When the cells reached 70%–80% coating, they were treated with trypsin-EDTA and then examined under a microscope. After homogenization, the cells were painted trypan blue and counted using a hemacytometer. The cells were then plated and incubated in a drying oven for 24 h.

The flasks were taken from the incubator and placed in a sterile environment in a laminar flow cabinet. After discarding the medium, 0.5 ml trypsin-EDTA was added to the flask and incubated for 2–3 min. After incubation, cells were examined under a microscope for separation from the flask surface. After adding flask media, the cells were transferred to a falcon tube and centrifuged for 2 min at 2,500 rpm. After centrifugation, the supernatant was discarded and the cell pellet was suspended.

Characterization of the bPEI/AuNP-based delivery system

This study used a modified laser ablation in liquid (LAL) approach to create AuNPs. The laser beams focused on the gold plate (99.99%) in PEI solution (5% wt in H2O) at room temperature (RT). A glass vessel was filled with 10 mL bPEI solution, and the gold plate was added. The Nd:YAG laser was used in this study. The first harmonic of this laser produces 1,064 nm wavelength light pulses. The Nd:YAG was operated at 1,000 mJoule and a frequency of 10 Hz for this study. The four mm diameter laser beam was focused on the gold target for five minutes through a convex lens. The redness of the solution gradually increased with the ablation time because of the nanoparticle formation. The hydrodynamic size of the supplied AuPEI NP was 25 nm, the PDI value was measured as 0.62, and the zeta potential value was −0.91 mV. Our earlier study included a detailed report on the synthesis and characterization of the nanoparticles (Çağdaş Tunalıet al., 2023; Dong et al., 2020). Gold nanoparticles were synthesized after a 3-hour incubation at 30 °C. The initial concentration was 0.5 mg/ml, which was then diluted by 50% to 0.25 mg/ml and then 0.125 mg/ml, and analyzed using these three concentrations (Çağdaş Tunalıet al., 2023; Dong, Tong & Qi, 2013; Dong et al., 2020).

The integration of siRNA with synthesized AuPEI

A total of 1 µL of siRNA (in concentrations of 25 nM, 50 nM, and 100 nM) was combined with 10 mg/mL of conjugate solution, then incubated for one hour at room temperature. Gel retardation analysis was used to confirm the complexes that were generated. After loading the siRNA and conjugate/siRNA complexes onto 1.7% agarose gel wells containing ethidium bromide, electrophoresis was carried out at 5 V/cm.

Internalization of gold nanoparticle conjugates/siRNA complexes

CAPAN-1 cells, at a density of 30,000 cells per well, were put in a 48-well plate and left to incubate overnight. The next day, the cell medium was discarded, and the cells were cultured for 24 h with a novel medium (OPTI-MEM®) containing precise quantities of complexes. Afterwards, the medium was removed and the wells were washed with PBS (1X, 10 mM) three times. In order to repair the cells, a sequential application of a 5% glutaraldehyde solution was used for 15 min at a temperature of 4 °C. The cells were then washed three times with PBS.

Preparation of dual dye solution and determination of apoptosis/ necrosis

Apoptosis and necrosis in cells were examined by double-staining cells with a ribonuclease A, Hoechst 33342, and PI solution. PBS was used in a poorly illuminated environment. Pre-prepared 48-well plates and 70 µl of staining solution were used for cell extraction. The plates were incubated for 15 min, and a fluorescent microscope (FITC/DAPI filter; Leica DM, Wetzlar, Germany) was used to evaluate the cells.

Cell viability analysis with the WST-1 assay

The WST-1 assay was used for analyzing cell viability. CAPAN-1 pancreatic cancer cells in flasks containing IMDM, L-glutamine, 20% FBS, and 1% antibiotic were incubated in a CO2 incubator for 48 h, then treated with AuPEI particles. After 24 h, the WST1 colorimetric assay procedure was performed, adding 5 µL of WST-1 reagent to each well. The plates were read in an ELISA Microplate Reader at a wavelength of 438 nm after an additional 4 h of incubation.

RT-PCR analysis

The real-time PCR experiment involved mixing 25 nM of siRNA with nanoparticles at different doses. The siRNA/AuPEI complex was incubated for 30 min at ambient temperature. The medium containing 5 × 105 cells was extracted from 6-well plates, and prepared solutions were introduced into the wells. A full solution was added, and the liquids were mixed with the cells. The plates were incubated for 48 h before being removed from the incubator.

The RNA was isolated from the cells using a High Pure RNA Isolation Kit (Roche, Basel, Switzerland), following the manufacturer’s instructions. Then, a cDNA synthesis was performed for each group using the Transcriptor High Fidelity cDNA Synthesis Kit, version 8.

The Hydrolysis Probes Master Kit, version 9, was used on the LightCycler 480 to determine the expression level of the target gene relative to other genes.

Agarose gel electrophoresis

The real-time PCR findings were validated by subjecting them to agarose gel electrophoresis. A gel with a concentration of 1.7% was produced for the RT-PCR products. A total of 10 µl, consisting of 7 µl of the PCR product and 3 µl of loading buffer, were injected into the wells. The samples were then subjected to an electric field strength of 5 V/cm. The results were captured using a gel imaging system.

Xcelligence® real-time cell analysis (RTCA) system

An xCELLingence® RTCA device, using electrical impedance technology, was used to measure cell proliferation with high sensitivity and accuracy without the use of markers and physiological contact, based on the ionic concentration in each well and whether the cells were attached to the electrodes. The e-plate was kept in an oven for 10-15 min to reach the same temperature as the device. From the previously prepared cells, 5,000 µl of cells were inoculated into each well of the e-plate and 100 µl of medium was added to each well. The prepared plate was placed in the device and then moved to an oven for incubation. After 24 h, the medium was discarded, and siRNA, bPEI/AuNP, and siRNA-bPEI/AuNP solutions were added to the wells. The device was placed in the oven again and impedance measurements were taken at 10-minute intervals. The entire run time was set to 96 h.

Statistical analysis

MS Excel and SPSS 21.0 were used to analyze the data. As descriptive statistics, median, mean, standard deviation, and minimum and maximum values were calculated. A normality test of the data was performed using the Shapiro–Wilk’s test. The results showed the data were not normally distributed (p < 0.05). Therefore, the Mann–Whitney U test was used to compare two groups, and the Kruskal-Wallis test was used to compare multiple groups. P < 0.05 was used as the significance level in all tests.

RESULTS AND DISCUSSION

AuPEI-based siRNA delivery system

To determine the optimal siRNA concentration, three different concentrations were tested: 100 nM, 50 nM, and 25 nM. Optimal values of the conjugate solution were determined via gel retardation analysis (1.7%, 5V/cm). The 25 nM siRNA concentration used in similar studies was shown to be the most effective (Yang et al., 2013; Bartlett & Davis, 2007). Gel images of siRNA and siRNA-bPEI/AuNP complexes are shown in Fig. 1. The agarose gel electrophoresis map shows the different siRNA loading groups with three amounts of PEI-AuNP/siRNA: 0.5, 0.25, and 0.125 µg/µl. The weight ratio of siRNA to PEI-AuNP was 1:1. The results showed a 0.5 µg PEI/AuNP-siRNA band in groups 1–6, with the siRNA band disappearing in groups 7–12. This result indicates that in vitro transfection performed better in the conjugation of siRNA with 0.25 µg/µl PEI/AuNP and 0.125 µg/µl PEI/AuNP than with 0.5 µg PEI/AuNP-siRNA.

Figure 1 An agarose gel electrophoresis image used to determine the optimal concentration of siRNA and complexes.

Three different amounts of PEI-AuNP/siRNA (0.5, 0.25 and 0.125 µg/µl) and siRNA (25 nM): the weight ratio of siRNA to PEI-AuNP was 1:1. Gel images of control (1–4), 0.5 µg PEI/AuNP-siRNA (5–6), 0.25 µg PEI/AuNP-siRNA (7–8), 0.125 µg PEI/AuNP-siRNA (9–10) and siRNA (11–12).

Modified laser ablation in liquid (LAL) was used to obtain PEI-coated gold nanoparticles, which were then used to effectively deliver siRNA. Gold nanoparticles are a reliable carrier system due to the non-toxic nature of siRNA and its ability to produce small nanoparticles. These nanoparticles can be modified using covalent or electrostatic techniques to increase gene delivery. The use of free thiol groups can create target molecules or stimulants for the release of siRNA. Adding a polyetherimide (PEI) layer to a gold nanoparticle (AuNP) core strengthens the connection with siRNA, which leads to a significant decrease in gene expression and an increase in cell death. The stimulating reactive behavior of gold nanoparticles allows siRNA molecules to be transported to the cytoplasm, effectively suppressing the target gene with minimal side effects (Song et al., 2010; Lee et al., 2008).

Apoptosis and necrosis are induced in CAPAN-1 cells by the siRNA KRAS

The apoptotic and necrotic mechanisms of the cells were evaluated using a Leica inverted fluorescence microscope after PI staining (FITC/DAPI filter; Leica DM6000, at 200X magnification). Images for the control and siRNA groups are shown in Fig. 2. The images obtained for the 0.5, 0.25, and 0.125 µg/µl PEI-AuNP/siRNA groups are shown in Fig. 3.

Figure 2 Apoptotic necrotic cell images obtained using Hoechst 33342 and PI fluorescent dye (Images were taken with a Leica inverted fluorescence microscope at 200X magnification).

(A) Control group CAPAN-1 cells (stained with Hoechst 33342); (B) control group CAPAN-1 cells (stained with PI); (D) CAPAN-1 cells treated with 25 nM siRNA (stained with Hoechst 33342); (E) CAPAN-1 cells treated with 25 nM siRNA (stained with PI). Arrows indicate some of the apoptotic and necrotic cells (BAR = 50 µm).

Figure 3 Apoptotic and necrotic cell images were obtained using Hoechst 33342 and PI fluorescent dye (images were taken with a Leica inverted fluorescence microscope at 200X magnification).

(A) CAPAN-1 cells treated with 25 nM siRNA/0.5 mg/ml bPEI/AuNP complex (stained with Hoechst 33342); (B) CAPAN-1 cells treated with 25 nM siRNA/0.25 mg/ml bPEI/AuNP complex (stained with Hoechst 33342); (C) CAPAN-1 cells treated with 25 nM siRNA/0.125 mg/ml bPEI/AuNP complex (stained with Hoechst 33342); (D) CAPAN-1 cells treated with 25 nM siRNA/0.5 mg/ml bPEI/AuNP complex (stained with Hoechst 33342 and PI); (E) CAPAN-1 cells treated with 25 nM siRNA/0.25 mg/ml bPEI/AuNP complex (stained with PI); (F) CAPAN-1 cells treated with 25 nM siRNA/0.125 mg/ml bPEI/AuNP complex (stained with PI). Cell nuclei that have undergone apoptosis appear bright and fragmented, while those that have not undergone apoptosis appear pale blue. Necrosed cell nuclei appear red. Arrows indicate some of the apoptotic and necrotic cells (BAR = 50 µm).

Apoptosis was induced by suppressing the KRAS gene through the application of siRNA-PEI/AuNP conjugate to CAPAN-1 cells. It was observed that the nuclei of apoptotic cells were disintegrated, and these cells were brighter blue than cells that did not undergo apoptosis (Figs. 2 and 3). There was no morphological difference in the cell nuclei of the control group compared with the nuclei of siRNA-treated cells (Fig. 2). These results indicate that siRNA is not cytotoxic.

Calculated apoptotic and necrotic index values are shown in Table 1. The data showed that the apoptotic effect of nanoparticles applied at low doses on the cells increased when given to CAPAN-1 cells together with 25 nM siRNA. This result indicates that KRAS contributes to the suppression mechanism. Results of the necrosis analysis demonstrated that siRNA also triggered the necrotic mechanism. Double staining results showed that viability decreased in cells treated with the siRNA complex. These results indicate that some cell deaths occurred via the apoptotic pathway and some via the necrotic pathway. The apoptotic index of cells interacting with 0.125 mg/mL PEI-AuNP/siRNA was higher than 0.25 mg/mL PEI-AuNP/siRNA and 0.5 mg/mL PEI-AuNP/siRNA (33.36 ± 16.421%, P < 0.05). The group with the highest necrotic index was 0.25 mg/mL AuPEI (19.6 ± 8.856%, P < 0.05).

Table 1 The control and treatment groups were examined for apoptotic and necrotic index values.

Group	Apoptotic index (%)	Necrotic index (%)	
Control	0,00	2,10	
siRNA	0,10	1,00	
siRNA/0,5 mg/mL AuPEI	29,16	17,16	
siRNA/0,25 mg/mL AuPEI	26,60	19,60	
siRNA/0,125 mg/mL AuPEI	33,36	15,76	

Cell viability with the WST-1 assay

Cell viability was assessed in response to siRNA, particles, and the combination of particles and siRNA. The WST-1 test findings indicate that siRNA exhibited higher cytotoxicity towards siRNA/AuPEI cells than toward untreated cells. The siRNA/AuPEI cells in the application groups, assessed by the WST-1 technique, had a viability rate of over 90%, indicating no toxicity or side effects.

To regulate the high absorbance value of gold particles in the AuNP-applied groups, the average of the AuNP dosages was subtracted from the siRNA-PEI/AuNP values, resulting in adjusted siRNA values. No statistical difference was observed between these values and the original siRNA values (p > 0.05; Table 2). AuNP absorbance seemed to be the reason why the cell viability in the AuNP groups appeared to increase compared to the control group (Fig. 4).

Table 2 Statistical evaluation between siRNA obtained from the study and original SiRNA (corrected) absorbance values (p > 0.05).

	SiRNA (adjusted)	SiRNA	Mann–Whitney U	
			Z	p	
Mean	0,21	0,24	−0.832	0,405	
Std deviation	0,12	0,01			

Figure 4 The cell viability was graphed as the relative percentage of cell viability for siRNA, particles, and combinations of particles and siRNA compared to the control group.

The standard deviation (±sd) is presented, and the statistical significance is shown as p < 0.05.

Cell viability of the groups in the study was evaluated by WST-1 analysis with an ELISA microplate reader. The absorbance values obtained at a 438 nm wavelength and the results of the statistical analysis (min., max., average, and standard deviation; p < 0.05) are shown in Fig. 5.

Figure 5 Absorbance values (438 nm) obtained on the ELISA microplate reader in the WST-1 analysis of the groups participating in the study and statistical analysis evaluation (min., max., mean, and std. deviation; p < 0.05).

The use of the xCELLigence® (RTCA) systems to carry out a cell proliferation test

With the xCELLigence RTCA® device, application groups were evaluated in the range of 0-97. xCELLigence RTCA® proliferation results of the control and siRNA groups were similar. Groups were evaluated at hours 20, 48, 72, and 96. No significant difference was observed between groups at hour 20. At 48 h, there was a significant difference in proliferation in the 0.25 mg/mL PEI/AuNP-siRNA application group compared to the control and siRNA groups (p < 0.05). At 72 h, a significant difference in proliferation was observed in the 0.5 mg/mL PEI/AuNP-siRNA, 0.25 mg/mL PEI/AuNP-siRNA, and 0.125 mg/mL PEI/AuNP-siRNA application groups compared to the control and siRNA groups. (p < 0.05). Similarly, at 96 h, all three groups were statistically different from the control and siRNA groups in terms of proliferation (p < 0.05). The statistical evaluation results of the cell index values of the application groups in the study for hours 20, 48, 72, and 96 are shown in Fig. 6. The graphs created using these values are shown in Fig. 7.

Figure 6 The statistical evaluation of the cell index values of the application groups in the study for the 20th, 48th, 72nd, and 96th hours.

Figure 7 Graphs obtained using the cell index values of the study’s application groups at hours 20 (A), 48 (B), 72 (C), and 96 (D).

Evaluation of the efficacy of gene silencing with in vitro RT-PCR

Real-time PCR analysis was conducted to assess KRAS gene expression. First, mRNA isolation was performed on CAPAN-1 cells obtained from the applications. Then, cDNA synthesis was performed on the isolated RNA samples. The resulting cDNA samples were amplified using synthetic primer sequences designed for the GD12 mutation region of the KRAS gene with a LightCycler480® analyzer. The relative data obtained was then evaluated. RT-PCR products were run on 1.7% agarose gel electrophoresis, and band profiles were evaluated to confirm the accuracy of the results. The CP values obtained from the amplification procedure conducted in the LightCycler480® analyzer were 28.88–35.98 for the housekeeping gene. The ACTB housekeeping gene data were within the expected range in all samples. However, the targeted region did not amplify siRNA, AuPEI/siRNA, and NK. Electrophoresis application resulted in band profiles in the 200-250 bp range for the housekeeping gene group. These results are shown in Fig. 8.

Recent studies have demonstrated that the nanoparticle/siRNA complex can be effectively used for gene silencing a target gene. Kamian et al. (2023) showed band profiles in individuals with and without the KRAS mutation. Strand et al. (2019) reported that the peptide-based NP is rapidly taken up by cancer cells in vitro, delivers KRAS-specific siRNA, inhibits KRAS expression, and reduces cell viability. They also demonstrated that this system can deliver siRNA to the tumor microenvironment, reduce KRAS expression, and inhibit pancreatic cancer growth in vivo. In another study, Mendt et al. (2018) reported that novel KRAS-siRNA-based nanotherapeutics have been developed by incorporating G12D mutated KRAS-siRNA into iExosomes, which are extracellular nanovesicles derived from human foreskin fibroblast-like mesenchymal stem cells (MSC) functionalized with CD47. These iExosomes exhibit high circulatory retention time and efficient cellular internalization via enhanced macropinocytosis in cancerous pancreatic cells with oncogenic RAS, specifically targeting cancer cells harboring the oncogenic KRAS G12D mutation (Vakhshiteh, Atyabi & Ostad, 2019; Whiteside, 2017). Additionally, preclinical studies have shown that KRAS siRNA-engineered iExosomes decrease KRAS expression in mice, leading to suppressed cancer cell proliferation, enhanced apoptosis, inhibited metastasis, and increased overall survival without causing cytotoxic effects (Mendt et al., 2018; Kamerkar et al., 2017).

Figure 8 The agarose gel electrophoresis photograph of the products obtained in RT-PCR analysis.

The present study demonstrates that siRNA and AuPEI effectively target the KRAS gene, inducing apoptosis and necrosis in cancer cells. Further research is required to elucidate the efficacy of lower siRNA doses, conduct a detailed analysis of intracellular mechanisms, perform clinical safety testing, ascertain the applicability of this method to other cancer types, investigate the effect of nanoparticle modifications, and examine multiple targeting strategies in combination therapies. These additional studies may enhance the clinical potential of this method.

CONCLUSIONS

The results of this study indicate that siRNA-mediated gene silencing can be used therapeutically. The study revealed that the gold nanoparticle/siKRAS complex with PEI effectively inhibited the proliferation of CAPAN-1 cells and efficiently silenced the expression of the KRAS protein in vitro. The results of this study are important for the advancement of new therapies for pancreatic cancer, but more studies are needed to validate these results.

Supplemental Information

Supplemental Information 1 Real-Time PCR data

The cell index values for the graphs given for cell proliferation.

Supplemental Information 2 Cell viability

The data showing the Cp values and amplification obtained on the LightCycler 480 device.

Supplemental Information 3 xCELLigence RTCA® data

This article was produced from a master’s thesis (Küçükekmekci, 2019).

Additional Information and Declarations

Competing Interests

Author Contributions

Data Availability

The authors declare that there are no competing interests.

Büşra Küçükekmekci conceived and designed the experiments, performed the experiments, prepared figures and/or tables, and approved the final draft.

Fatma Azize Budak Yıldıran conceived and designed the experiments, performed the experiments, analyzed the data, prepared figures and/or tables, authored or reviewed drafts of the article, and approved the final draft.

The following information was supplied regarding data availability:

The raw data are available in the Supplemental Files.

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
