# Peer review of "Investigation of the efficacy of siRNA-mediated KRAS gene silencing in pancreatic cancer therapy"

_PeerJ, doi:10.7717/peerj.18214_

## Round 0.1 · original submission · Major Revisions

This manuscript needs major revisions primarily due to the following issues:

Basic Reporting
The manuscript contains numerous grammatical errors and awkward phrasing, affecting readability. A thorough proofreading is necessary to improve the overall quality.

The background section is overly extensive and lacks focus. Authors should provide a concise overview of the research context and clearly state the rationale for their study.

Figures need modification to align better with the study's objectives and to convey scientific information more effectively. Clear, relevant figures directly supporting the study's findings are essential.

The introduction is lengthy and would benefit from better organization and a more concise presentation of background information, possibly condensing details about different cancer treatment methods into a separate section.

Experimental Design
The rationale for using nanoparticles in conjunction with siRNA is not well-defined. Without a clear methodology or validation, this combination remains scientifically unsubstantiated.

Specific details in the materials and methods section are lacking, which could hinder reproducibility. Detailed information on nanoparticle synthesis, characterization, siRNA transfection, and cell culture conditions must be provided.

The results section requires better organization and clearer separation of experimental findings. Figures 2 and 3 lack clear captions or explanations, making data interpretation difficult.

The siRNA transfection process needs detailed description, including transfection reagents, cell lines used, and experimental conditions.
Evidence demonstrating nanoparticle internalization and functional siRNA retention is lacking, which is critical to support the study's conclusions.

Validity of the Findings
Without addressing the above points, the study's robustness for publication is in question. Additional experiments or revisions are necessary to strengthen the findings.

The statistical analysis section is brief and lacks specific details on tests performed and criteria for determining significance. Proper statistical methods and corrections for multiple comparisons should be employed.

Additional Comments
Unnecessary tables and figures should be removed or revised. Figures lacking quantification data or clear labels need improvement.
The manuscript presents interesting findings on siRNA-mediated gene silencing but requires significant enhancements to its scientific quality, clarity, and overall impact for publication.

In summary, substantial revisions are needed to address these fundamental issues and improve the manuscript's scientific quality, coherence, and impact potential. Addressing these concerns will enhance the clarity and reliability of the study's findings.

**Language Note:** The Academic Editor has identified that the English language must be improved. PeerJ can provide language editing services - please contact us at [email protected] for pricing (be sure to provide your manuscript number and title). Alternatively, you should make your own arrangements to improve the language quality and provide details in your response letter. – PeerJ Staff

Reviewer 1 ·

Basic reporting

The manuscript contains grammatical errors and awkward phrasing. A thorough proofreading is necessary to enhance readability.
The background section is overly extensive and lacks focus. Authors should provide a concise overview of the research context and clearly state the rationale for their study.
A succinct background will help readers understand the significance of the research without unnecessary details.
Figures play a crucial role in conveying scientific information. However, the current figures need modification to align with the research study.
Authors should ensure that figures are relevant, clear, and directly support the study’s findings.

Experimental design

The article lacks a well-defined rationale for using nanoparticles in conjunction with siRNA.
Without proper validation or methodology, the combination of nanoparticles and siRNA remains scientifically unsubstantiated.
Authors should explicitly state whether they aim to study the stability of siRNA when complexed with nanoparticles or focus on efficient delivery.

The article does not adequately describe the siRNA transfection process.
Authors should provide details on the transfection method used, including the choice of transfection reagents, cell lines, and experimental conditions.
The study lacks data demonstrating nanoparticle internalization and the retention of functional siRNA.
To strengthen the research, authors must present evidence of successful nanoparticle uptake by cells and subsequent siRNA delivery.

Validity of the findings

Without addressing the above points, the study may not be robust enough for a separate publication.
Authors should consider additional experiments or revisions to strengthen their findings.

Additional comments

Overall, the article requires substantial revisions to enhance its scientific quality. Addressing the mentioned issues will significantly improve its overall impact and suitability for publication.
Table 1 & 3 are unnecessary
Figure 1 - Figure labels are not to the level of research publication.
Figure 3& 4 need quantification data

Reviewer 2 ·

Basic reporting

1. The introduction section is quite lengthy and could benefit from better organization and concise presentation of the background information.
2. Some of the information, such as the details about different cancer treatment methods, could be condensed or moved to a separate section.

Experimental design

3. The materials and methods section is lacking specific details in certain areas, which may hinder the reproducibility of the study.
4. The synthesis and characterization of the gold nanoparticles (AuNPs) are briefly mentioned, but more detailed information on the synthesis method, size, and other relevant properties of the AuNPs should be provided.
5. The concentration ranges and ratios of siRNA and AuNPs used in the experiments should be clearly specified.
6.The cell culture conditions, such as the number of passages or the seeding density, are not mentioned.
7. The results section could benefit from better organization and clear separation of the different experimental findings.
8. Figures 2 and 3 lack clear captions or explanations, making it difficult to interpret the data. Some figures are with tools interface, recomended to remove the tools interphase present xCELLigence Figure 9. On ecan export the data and genrate their own figures.

Validity of the findings

10. The statistical analysis section is brief and lacks details on the specific statistical tests performed and the criteria used for determining significance
11. It is unclear if the reported significant differences are based on appropriate statistical tests and corrections for multiple comparisons.

Additional comments

The manuscript presents interesting findings on siRNA-mediated KRAS gene silencing, it requires additional details, clarifications, and improvements in various sections to enhance its scientific quality and clarity.

---

## Round 0.2 · Minor Revisions

The manuscript improved but the authors should include a discussion for the results.

Reviewer 1 ·

Basic reporting

This review addressed major comments

Experimental design

The experimental design is greatly improved and is acceptable.

Validity of the findings

Accetable

Additional comments

The authors should include a discussion for the results.

---

## Round 0.3 · accepted · Accept

Authors have addressed all reviewer comments.